# Analysis of an Adaptive Periodic Low-Frequency Wave Filter Featuring Magnetorheological Elastomers

**DOI:** 10.3390/polym15030735

**Published:** 2023-01-31

**Authors:** Hamid Jafari, Ramin Sedaghati

**Affiliations:** Department of Mechanical, Industrial and Aerospace Engineering, Concordia University, Montréal, QC H3G 1M8, Canada

**Keywords:** adaptive metamaterials, magnetorheological elastomers (MREs), wave propagation, wave filters, periodic structures

## Abstract

This study aims to enhance and tune wave-propagation properties (Bandgaps) of periodic structures featuring magnetorheological elastomers (MREs). For this purpose, first, a basic model of periodic structures (square unit cell with cross-shaped arms), which does not possess noise filtering properties in the conventional configuration, is considered. A passive attenuation zone is then proposed by adding a cylindrical core mass to the center of the conventional geometry and changing arm angles, which permitted new bandgap areas. It was shown that better wave-filtering performance may be achieved by introducing a large radius of the cylindrical core as well as low negative cross-arm angles. The modified configuration of the unit cell was subsequently utilized as the basic model for the development of magnetoactive metamaterial using a MRE capable of varying the bandgaps areas upon application of an external magnetic field. The finite element model of the proposed MRE-based periodic unit cell was developed, and the Bloch theorem was employed to systematically investigate the ability of the proposed adaptive periotic structure to attenuate low-frequency noise and vibration. Results show that the proposed MRE-based periodic wave filter can provide wide bandgap areas which can be adaptively changed and tuned using the applied magnetic field. The findings in this study can provide an essential guide for the development of novel adaptive periodic structures to filter low-frequency noises in the wide frequency band.

## 1. Introduction

Architected metamaterials and phononic crystals are novel manufactured materials with textile geometries. They demonstrate specific behaviours (e.g., negative stiffness, mass, permittivity, refractive index) not attended in conventional materials [1], making them favourable nominees for various applications like sensors [2], actuators [3], elastic/acoustic filters [4,5], soft robotics [6], and wearable electronics [7], etc. In acoustics, the periodic configurations in these metastructures are designed to resist the propagations of elastic/acoustic waves within specific frequency ranges (or stopbands, known as band gaps). These structures enable exciting applications such as sound cloaks [8], waveguides [9], and vibration filters [10] by properly designing the shape of unit cells (as the smallest repeating unit in different directions) with targeted frequency ranges. Although the research into architected metamaterials is considerable and rapidly increasing, there are still some limitations, such as the lack of low-frequency bandgaps and tunability. In fact, low-frequency noise and vibration bring new challenges in newly developed technologies [7], and the design of architected structures to control these harmful vibrations from the propagation pathway over the structural design has become a significant research topic [11,12,13]. In addition, clear links have been established between excessive exposure to industrial noise and adverse health effects, such as hearing loss, pain, disrupted sleep cycles, cardiovascular disorders, and impaired cognitive development of workers and labourers in transport and construction, natural resources, and related production occupations [14,15]. As an environmental stressor, noise negatively impacts psychological and physical well-being [16]. Accordingly, developing novel methodologies and advanced materials to reduce the detrimental effects of noise is of paramount importance for human well-being [17].

In order to deter and filter low-frequency noise and vibration, the current research in architected periodic structures mainly concentrates on passive design and active control. The architected metamaterials’ development by considering the symmetry of the unit cells, the selection of desired materials, and the array of structural elements satisfies the passive design of the architected metamaterials. However, most existing developments rely on architected microstructures, which are not tunable after fabrication; in fact, their properties cannot be modulated once the structures are realized [18]. In addition, the effectiveness of sound protectors comes from the sound attenuation they provide but depends on correct and continuous use throughout the exposure [19]. This last factor is significantly affected by the lack of comfort associated with the help of these protectors. There is a significant acoustic discomfort in using these ear protectors, i.e., the comfort associated with the intelligibility of external sounds [19], which causes workers not to use them continuously.

In recent years, some researchers [5,20,21,22] have obtained various methods to find structures with the desired wide band gaps, which can be tuned with external stimuli. Yang et al. [23] employed thermal-sensitive materials to tune band gaps and waveguiding properties with external actuation. They could effectively manipulate the width and location of band gaps and develop an on-off wave switch with thermal excitations. Besides, Sepehri et al. [24] used piezoelectric springs in the one-dimensional periodic structure to tune various types of band gaps (Local Resonance and Bragg-type) with electrical stimuli, which caused a change in the elastic modulus of resonators and their natural frequencies. The mentioned methods and many others, like chemo-active [25], light-responsive [26], and magneto-active approaches, were used to manipulate elastic or acoustic properties of architected structures; however, most of them show specific restrictions in their development. For instance, engineers need direct contact with the architected structure to stimulate electroactivity—the opacity of the surrounding media limits the light-responsivity of materials. Thermo-activity and chemo-sensitivity of materials require local environmental conditions. Using each method has approachability issues in the structural design, but magneto-active properties have significant advantages in tuning properties. The main benefits of using this property of materials are non-contact (wireless) activation, fast switching, nonlocal motivation, and very low power requirement [27].

Magneto-responsive or magneto-active materials (MAEs) are smart functional materials that change their architecture and/or mechanical properties (like elastic modulus) under external magnetic stimulations. Recently, these materials have been widely used to develop and fabricate structures that need a fast response to external stimuli or adaptivity in their desired application, e.g., reconfigurable morphing architectures [28] for soft robots [29] and drug delivery structures [30], multifunctional magnetic materials [31] for sensors [32], and actuators [33]. Using these materials as a foundation of periodic architected structures may significantly help to develop tunable elastic/acoustic filters, which can have various biomedical and industrial applications. Zou et al. [21] used permanent magnets in the corners of the square lattice made with shape memory polymers to develop adaptive Achiral and Chiral structures. They demonstrate the mutual assistance of two physics concepts—magnetic control combined with the thermomechanical behaviour of shape memory polymers to control band gap areas with magnetic and thermal stimuli. Additionally, Montgomery et al. [31] designed and fabricated magneto-mechanical metamaterials with the shape-morphing concept and asymmetric joint design. The hard-magnetic soft active materials are used to enable two distinct actuation modes (bending and folding) under opposite-direction magnetic fields. Their magnetized structures are able to change mechanical and wave propagation properties with external magnetic fields. The main focus of previously studied research was on shape morphing and adaptivity based on initial magnetization in the periodic structures, which may not be applicable in applications like tunable industrial noise filters, hearing protectors, and vibration isolators. Magnetorheological elastomers (MREs) as a primary material of periodic structures have become recently more significant based on their unique adaptive mechanical properties under an external magnetic field [34]. 

MREs are unique multifunctional materials capable of changing their dynamic properties (stiffness and damping) instantly (milliseconds) under the application of an external magnetic field. Moreover, due to their inherent magnetostriction property, they can provide magnetically driven soft actuation. The hard and soft magnetizable particles may be distributed consistently (isotropic) or in a chain-like manner (anisotropic) within the matrix [35]. The mechanical characteristics of MREs under different operational modes have been extensively studied in recent years to understand their response behaviour under varying mechanical and magnetic loadings [36]. Thanks to their rapid response and low power consumption, they present attractive capacities for numerous engineering applications, especially tunable filters. Xu and Wu [37] used MREs in a multi-layered one-dimensional phononic crystal to isolate elastic vibration adaptively. Xu et al. [38] later considered tungsten as the core of MRE material inside the epoxy materials to create a unit cell for a 3D periodic structure (tungsten as mass, epoxy as the matrix, and MRE as cladding). They showed that the external magnetic field is capable of controlling the elastic modulus or thickness of the MRE cladding to change the band gap position and width for isolation bearings. Additionally, Pierce et al. [39] introduced a 3D printing method to fabricate and test three-dimensional MREs for adaptive vibration filters. They concluded that the band gap tunability depends not only on the strength of the applied magnetic field but also on the metastructure geometry. Zhang and Rudykh [40] studied the transverse elastic waves in 1D hard-magnetic soft laminates. They employed a small-on-large framework and examined small-amplitude shear wave propagation in the magnetoactive structure. They found a compressive deformation in the applied magnetic field direction, which caused a change in wave propagation behaviours. Moreover, Zhang et al. [41] recently presented asymmetric mechanical metamaterial that integrated hard-magnetic elastomers into designing metastructure that considers the bi-stability, local resonant, and snap-through effects. They showed how the magnetic field affects material stiffening and changing the elastic wave propagation properties in their specific configuration of metamaterials.

In the present study, a MRE-based square lattice will be initially used as a unit cell to design magnetoactive architected periodic metamaterial, which is one of the primary 2D unit cells presented by Phani et al. [42]. This unit cell is commonly utilized as a primary form for periodic structures in a number of research studies, with no bandgap area in low-frequency vibrations. A cylindrical core and trapezoidal arms will be subsequently introduced into a square lattice to enhance bandgap regions. The band gap adaptability of the proposed architected structures featuring magnetorheological elastomers with various iron volume fractions under varied applied magnetic fields will be investigated. The evolution of the bandgap and filtering area in the absence (passive) and presence of the applied magnetic field will be presented. Finally, the relation between the bandgap area and the applied magnetic flux density will be discussed.

## 2. Materials and Methods

Unit cell geometrical parameters play an important role in the dynamic behaviour and wave-propagation properties of periodic structures. It has been shown that the conventional square lattice unit cell (Figure 1a) is not able to attain a bandgap area in low-frequency vibrations, partly due to mean node connectivity [43] and balanced mass spread in all areas of the unit cell [44]. As represented by Sepehri et al. [43], the mean connectivity issue may be addressed by adding more beams and nodes to the basic unit cells due to the hierarchy of conventional structures like square and hexagon. However, this poses some limitations in the fabrication of these metamaterials. Here, in this study, distribution of the mass inside the MRE-based unit cell has been altered to develop adaptive periodic structures (with variable bandgap areas). Figure 1b shows the proposed MRE-based cross-shaped unit cell, including a cylindrical core and trapezoidal arms. Important geometrical parameters affecting bandgap areas are shown in this figure. The dynamic properties of the proposed adaptive unit cell can be effectively varied using the magnetic stimuli shown in Figure 1c.

### 2.1. Theoretical Model of Magnetic Effect

Magnetorheological elastomers with silicone rubber (Mold Max, Smooth-on) as the host matrix impregnated with different volume fractions of magnetic iron particles are considered to develop periodic architected structures. By applying the magnetic field perpendicularly to the plane of the MRE unit cell structure, the governing equation of motion for the magneto-structural system can be generally presented in finite element form as [45]:(1)$${\bar{M}}^{s}\ddot{\vec{u}}+\bar{K}{}^{s}\vec{u}=\vec{F}{}_{EX}^{s}+\vec{F}{}_{M}^{S},$$
where $\bar{M}{}^{s}$ and $\bar{K}{}^{s}$ represent the mass and stiffness matrices of the unit cell, respectively. Additionally, $\vec{F}{}_{M}^{S}$ and $\vec{F}{}_{EX}^{s}$ describe the Maxwell force vector due to the applied external magnetic field and external load vector, respectively. For the harmonic excitation, the nodal displacement vector of the architected structures ($\vec{u}$) is a harmonic function, which can be described as: $\vec{u}=\vec{q}e^{i\omega t}$, in which $\vec{q}$ is the nodal displacement amplitude vector, and ω is the excitation frequency. For the free wave motion analysis, the external load vector ($\vec{F}{}_{EX}^{s}$) is zero, and the Maxwell force vector applied over the volume (*V*) can be written as [45]:(2)$$\vec{F}{}_{M}^{s}=\int_{V}{\bar{B}}^{T}\sigma_{M}dV,$$
where ${\bar{B}}^{T}$ is the strain-displacement matrix [46]; $\sigma_{M}$ is the Maxwell stress tensor, which can be calculated as [47]:(3)$$\sigma_{M}=B_{s}⊗H_{s}-\frac{1}{2}\left(B_{s}.H_{s}\right)I,$$
where $B_{s}$ and $H_{s}$ represent the magnetic flux density and imposed magnetic field intensity in the magnetoactive elastomer, respectively, and I is the identity tensor. It is noted that the magnetic flux density is related to the magnetic field intensity as:(4)$$B_{s}=\mu_{0}\left(1+\chi \right)H_{s},$$
where $\mu_{0}=4\pi \times 10^{-7}N/A^{2}$ is the permeability in the vacuum, and χ describes the magnetic susceptibility. 

Considering the Equations (2)–(4) in Equation (1), with the assumption of the Neo-Hookean hyperelastic material model for a nearly incompressible magnetoelastic medium [48,49] and linear elastic wave motion [50,51], and also considering field-dependent material properties of magnetorheological elastomers [49], the governing equation of motion for a unit cell in the presence of induced magnetic flux density, B, can be evaluated as:(5)$$\left[\bar{K}\left(B\right)-\omega^{2}\bar{M}\left(B\right)\right]\vec{q}e^{i\omega t}=\vec{F}{}_{M}^{s},$$
where the $\bar{K}\left(B\right)$ and $\bar{M}\left(B\right)$ are the field-dependent mass and stiffness matrices of the unit cell, respectively, which are dependent on external magnetic flux (B), elastic modulus (E), unit cell geometry, density, and magnetic susceptibility (χ) of the filled elastomer with a specific amount of iron volume fraction.

In order to solve Equation (5) and find the frequency of ω in wave propagation analysis, it is essential to define the appropriate periodic boundary condition for each specific unit cell and reduce the order of the equation of motion to solve it for the architected periodic structure. 

### 2.2. Bloch Periodic Boundary Conditions

In periodic structures based on the concept of Bravais lattice, one can effectively select and analyze a reference unit cell and then generalize the results to all its near and far neighbour cells using some periodicity-based procedures such as the Bloch theorem [52]. This approach will drastically reduce the computational time. According to this concept, the displacement of the points inside any unit cell in a periodic structure is a function of the displacement of the corresponding points in the reference unit cell. Figure 2a shows a sample structure of the square model, its unit cell, and direct vectors $\left({\vec{a}}_{1},{\vec{a}}_{2}\right)$. Considering the reference unit cell, integers $m_{1}$ and $m_{2}$ are used in ${\vec{a}}_{1}$ and ${\vec{a}}_{2}$ directions to define any other unit cells. The position vector of a point inside the cell of $\left(m_{1},m_{2}\right)$, i.e., $\vec{R}$, relates to the position vector of the corresponding point in the reference unit cell, ${\vec{R}}_{i}$, using:(6)$$\vec{R}={\vec{R}}_{i}+m_{1}{\vec{a}}_{1}+m_{2}{\vec{a}}_{2};\;\;\;\;m_{1},m_{2}=0,\pm 1,\pm 2,\ldots \;,$$

Based on the Bloch theorem, the displacement of any point in a specified unit cell can be described as:(7)$$\begin{array}{c} \vec{q}\left(\vec{R}\right)=\vec{q}\left({\vec{R}}_{i}\right)e^{\vec{\kappa }.\left(\vec{R}-{\vec{R}}_{i}\right)}=\vec{q}\left({\vec{R}}_{i}\right)e^{\vec{\kappa }.\left(m_{1}{\vec{a}}_{1}+m_{2}{\vec{a}}_{2}\right)},\\ \{\begin{array}{c} k_{1}=\vec{\kappa }.{\vec{a}}_{1}\\ k_{2}=\vec{\kappa }.{\vec{a}}_{2} \end{array}⟹\vec{q}\left(\vec{R}\right)=\vec{q}\left({\vec{R}}_{i}\right)e^{\left(m_{1}k_{1}+m_{2}k_{2}\right)}, \end{array}$$
where κ is the wave vector, and $k_{1}$ and $k_{2}$ represent its components along the ${\vec{a}}_{1}$ and ${\vec{a}}_{2}$ directions, respectively. Additionally, $k_{1}=\vartheta_{1}+i\varepsilon_{1}$ and $k_{2}=\vartheta_{2}+i\varepsilon_{2}$ have real ($\vartheta_{i}$) and imaginary ($\varepsilon_{i}$) parts called the attenuation and phase constants, respectively. The real part is a measure of the attenuation of a wave as it progresses from one unit cell to the next, and the imaginary part or the phase constant is a measure of the phase change across one unit cell. For the sake of simplicity and waves propagating without damping alternation from a unit cell to others, in this study, the attenuation is assumed to be zero; thus, $k_{1}$ and $k_{2}$ are considered as $k_{1}=i\varepsilon_{1}$ and $k_{2}=i\varepsilon_{2}$.

In order to study wave propagation, the reciprocal space equivalent to direct space should be defined. The reciprocal vectors $\left({\vec{b}}_{1},{\vec{b}}_{2}\right)$ required to find the Brillouin zone are computed as:(8)$$\{\begin{array}{c} {\vec{b}}_{1}=\frac{2\pi }{\left|{\vec{a}}_{1}\times {\vec{a}}_{2}\right|}\left[\begin{array}{c} a_{2y}\\ -a_{2x} \end{array}\right]\\ {\vec{b}}_{2}=\frac{2\pi }{\left|{\vec{a}}_{1}\times {\vec{a}}_{2}\right|}\left[\begin{array}{c} -a_{1y}\\ a_{1x} \end{array}\right] \end{array}\;,\;{\vec{a}}_{3}=\left(0,0,1\right),$$
where $\left(a_{1x},a_{1y}\right)$, $\left(a_{2x},a_{2y}\right),$ and (0,0,1) are the components of the direct vectors ${\vec{a}}_{1}$, ${\vec{a}}_{2}$, and ${\vec{a}}_{3}$ along the x-, y-, and z- directions, respectively. Using reciprocal space, also called k-space, allows representing the Fourier transform of the spatial function (in real space) with respect to spatial frequencies (wave vector numbers $k_{1}$,$k_{2}$). One of the main parts of the wave propagation analysis is finding the relation between the free wave motion frequency and its associated wavenumbers ($k_{1}$,$k_{2}$), known as a dispersion relation. The dispersion relation should be obtained in the first Brillouin zone (FBZ), in which the Brillouin zone is the unit cell in the reciprocal space [53], as shown in Figure 2b. This allows to formulate the relation between the frequencies and wavenumber. The FBZ and IBZ (irreducible Brillouin zone) of the square unit cells in reciprocal space are shown in Figure 2b. It is noted that due to the symmetry of FBZ, the IBZ is the smallest domain necessary to solve the dispersion relation. Based on the Bloch theorem, the wave vectors are restricted to the edges of the IBZ, shown by the hatched region in Figure 2b. The high-symmetry points (G–X–M) are calculated from the reciprocal space vectors and FBZ area due to the symmetries of the primitive unit cell. Thus, the ranges of the phase constant (i.e., imaginary part of the wave vector) of $\{\begin{array}{c} \varepsilon_{1}\\ \varepsilon_{2} \end{array}$ are {0→b120, {b120→b12, and {b12→0b12→0 in the G–X, X–M, and M–G directions, respectively.

Substituting Equation (7) in Equation (5) and using Bloch–Floquet periodic boundary conditions (periodic boundary conditions for upper with lower and left with right sides of the unit cell), one can conduct wave propagation analysis in reciprocal (k-space) using the following governing equation:(9)$$\left[\bar{K}\left(\vec{\kappa },B\right)-\omega^{2}\bar{M}\left(\vec{\kappa },B\right)\right]\vec{q}e^{i\omega t}=\vec{F}{}_{M}^{s},$$
where the $\bar{K}\left(\vec{\kappa },B\right)$ and $\bar{M}\left(\vec{\kappa },B\right)$ are periodic stiffness and mass matrixes, respectively, related to magnetic flux (B) and reduced wave vector ($\vec{\kappa }$).

## 3. Results and Discussion

This section covers the results of the wave propagation analysis in the proposed adaptive architected periodic structures based on magnetorheological elastomers. The design parameters of the proposed modified unit cell (i.e., the radius of the cylindrical core, *R*, and the angle of the arms, *θ*, in the square unit cell in Figure 1b) will be initially fine-tuned to find a proper range of bandgap areas. The length, connecting edge of the arms, and thickness of the unit cell are considered to be *L* = 5 mm, *d* = 0.25 mm, and *t* = 0.2 mm, respectively. In addition, the material properties of the magnetorheological elastomers used in this study are presented in Table 1, which are experimentally measured by Kyung Hoon Lee et al. [54] for the magnetorheological elastomer fabricated using the mixture of silicone rubber (Mold Max, Smooth-on) and different iron volume fraction of magnetic particles. Table 1 provides Young’s modulus, density, and Poisson’s ratio, as well as the effective magnetic susceptibility difference, (Δχ=χ−χ/(1+χ/2)), for the different iron volume fractions of MREs. These material properties, together with geometrical parameters of the unit cell, are considered to develop a multi-physics model of the proposed adaptive MRE-based metastructure with periodic boundary conditions based on Equation (9) using COMSOL Multiphysics. It is noted that in the analysis, MRE material is assumed to be a hyperelastic material in which the effect of damping has been ignored due to its insignificant effect in free wave motion analysis and dispersion relation.

### 3.1. Passive Design (Field-off) of Architected Structures

To demonstrate the generation of band gap regions in the proposed periodic MRE-based metastructure in the absence of the magnetic field, the branch diagram for the conventional periodic square lattice (Figure 1a) and the proposed periodic structure with cylindrical core and trapezoidal arms (Figure 1c) are depicted in Figure 3. Results in Figure 3a clearly show that the conventional configuration is not able to show any specific stopband area in the low-frequency range of 1–5 kHz. Adding a cylindrical mass (with a ratio of L/R=5) at the core and considering angle of arms ($\theta =3^{{}^{\circ}}$), the structure develops band gap feature in various areas in this range of frequency, as shown in Figure 3b.

The generation of the new bandgaps is in-part due to the fact that the elements of the modified unit cell have natural frequencies within the specified range, causing local resonance bandgap areas. The conventional periodic model can be considered a periodic mass-spring model in two directions (*x*,*y*), with uniform constant mass distribution. Thus, by adding a cylindrical mass in the core of the unit cell and changing the arm angle, the mass distribution over the unit cell will no longer be constant. In order to verify added bandgap areas, the finite-period transmission spectrums are evaluated in Figure 1a,b for the conventional periodic square lattice and the proposed periodic structure with cylindrical core and trapezoidal arms, respectively. Results show a perfect agreement between bandgap areas (highlighted green areas) and sharp drops in the transmission spectrum. Amin et al. [44] also reported that variable mass distribution in the unit cell enables the development of bandgap areas (splitting specific branches in the wave propagation diagram). This can be further realized by comparing the sixth mode shapes of the conventional and modified lattices in various high symmetry points, as shown in Figure 3e,f, respectively. It is interesting to note that the proposed modified unit cell enables more band gap areas through the periodic model without changing the dimensions of the unit cell (i.e., ${\vec{a}}_{1},{\vec{a}}_{2}=cte$ or L,d=cte). This allows to design novel periodic structures with maximum bandgap regions. 

#### 3.1.1. Parametric Study on the Effect of Radius of Cylindrical Core in Passive MRE Unit Cell

The aim of the parametric study is to investigate the effect of geometrical parameters for the core radius, R, and arm angle, θ, as well as the material factor (the percentage of iron particles) on bandgap regions passively (in the absence of the magnetic field). The first part of the parametric design in the modified architected periodic structure is to alter the radius of the cylindrical core in order to find the desired unit cell for a specific range of frequencies. Figure 4 shows the results of bandgap regions within the 0−5 kHz frequency range for various radii of the core cylinder given arm angle θ=0 and iron volume fraction of 13.94%. As depicted in Figure 4a, the dispersion curve of the modified unit cell with the cylindrical core of L/R=5 has four different bandgap areas, which are mainly under 3 kHz. Results show that a periodic architected structure with this unit cell is able to filter vibrations in the frequency ranges of 0.70<ω<1.04 kHz, 1.25<ω<1.80 kHz, 2.33<ω<2.90 kHz, and 4.12<ω<4.16 kHz. By reducing the radius of core to L/R=8 without changing any other parameters, the first and last band gap areas are disappeared. The upper and lower branches of the second and third band gaps approach each other, as shown in Figure 4b. As the radius is further reduced to L/R=13, the dispersion curve becomes similar to that of convectional square lattice (i.e., Figure 3a) with a small bandgap region in higher frequencies (4.06<ω<4.19 KHz) due to the small radius of the cylindrical core. It is interesting to note that this small bandgap region for the unit cell with L/R=13 corresponds to the last band gap for the unit cell with the higher radius (L/R=5). This means that although the areas of these two bandgaps are approximately similar, the mode shapes associated with them in unit cells are different.

To better understand the effect of core radius, results for bandgap frequency range and percentage of covered bandgap areas within the frequency range 0−5 kHz are provided for L/R ratio ranging from 2 to 15, as shown in Figure 5a,b, respectively. As illustrated in Figure 5a, the modified periodic structure with different added cylindrical masses has various wave propagation behaviours. This provided an opportunity to generate a filtering area in multiple frequency ranges. Examination of results in Figure 5a reveals that filtering vibrations in the low-frequency range of 500–1800 Hz are achievable with periodic arrays of the square lattice with a relatively large radius (L/R<6). Results also show that periodic architected structures with lower radii (L/R≥13) are only able to have high-frequency filtering in frequencies above 4 kHz.

According to several parametric design studies [43,55] in wave propagation and bandgap analysis, it is important to analyze the coverage percentages of the filtering area. Figure 5b depicts the variation of bandgap coverage between the frequency range 0 to 5 kHz. As it can be realized from this figure, the maximum peak in percentage bandgap area (30%) occurs in nearly L/R=5. As a result, the periodic lattice with the radius of L/R=5 is selected for the modified unit cell. It is noted that, although selecting higher radius (lower L/R) may result slightly higher percentage of bandgap areas (ex L/R=2),  this is not desirable, as the amount of material used increases substantially. It is also noted that the unit cell with L/R=5 has band gap areas in low- and mid-level frequencies, which are appropriate for vibration-filtering devices.

#### 3.1.2. Parametric Study on the Effect of Radius of the Angle of Trapezoidal Arms in Passive MRE Unit Cell

The effect of the angle of trapezoidal arms, θ, on the bandgap areas has also been investigated for the proposed MRE unit cell, as shown in Figure 1b. The core radius is set at L/R=5 based on the discussion in Section 3.1.1, and the MRE with an iron volume fraction of 13.94% is chosen (Table 1). Results for unit cell angular frequency with respect to reduced wave vector, *k*, as well as its associate 6th model shapes for unit cell arm angles of$\;\theta =-1.5^{{}^{\circ}}\;,\;\;3.5^{{}^{\circ}},\;\mathrm{and}\;8^{{}^{\circ}}\;$are provided in Figure 6. Results show that metastructures with unit cells having lower angles of trapezoidal arms show more band gap areas in comparison to the structures with higher angles. As presented in Figure 6a, for instance, the unit cell with the angle of $\theta =-1.5^{{}^{\circ}}\;$has four filtering areas (mainly below 3 kHz) in the frequency ranges of 0.64–0.89 kHz, 1.03–1.74 kHz, 2.17–2.71 kHz, and 2.91–3.07 kHz. The bandgap areas reduce as the arm angle increases to $3.5^{{}^{\circ}}$ (three narrow band gaps) and substantial reduction occurs as the arm angle increases to $8^{{}^{\circ}}\;\left(\mathrm{one}\;\mathrm{narrow}\;\mathrm{band}\;\mathrm{area}\right).\;$

For further examination, let us focus on the first band gap on the basis of the local resonance phenomenon. As depicted in Figure 6d–f (i.e., mode shapes of the modified structure), the trapezoidal arms vibrate without any changes in the position of the cylindrical mass. This can also be realized from the steadiness of the branch on the upper side of the proposed band gap (Purple line, associated with the local 6th model shape). Except for the high symmetric point of M, which has phonon-phonon scattering because of veering in the dispersion curves, the rest of the branch is a straight line indicating the local resonance [56].

Changing the angle of trapezoidal arms affects the mass distribution along the unit cell. Low negative angle $\left(-1.5^{{}^{\circ}}\right)$ of the arms in the periodic model of the unit cells is more likely to be represented by the two lumped masses with a long distance between them represented by the soft spring, whereas for high positive angles $(+8^{{}^{\circ}}$) of the arm, based on the concentration of the mass distribution, it may be represented by three masses with shorter distances (stiff springs), as shown in Figure 7.

Results for the bandgap frequency ranges and percentage of band gap areas with respect to trapezoidal arm angle ranging from $\theta =-2.5^{{}^{\circ}}$ to $\theta =+10^{{}^{\circ}}$are shown in Figure 8a,b, respectively. Results show that the modified periodic structure with different angles of trapezoidal arms has various wave propagation behaviours. This dependency on the arm angle can be effectively used to design passive MRE-based period structures (field-off) capable of filtering vibration in specified frequency ranges. Figure 8a shows that the proposed MRE-based periodic structure, in the absence of the applied magnetic field, has an excellent low frequency (under 3 kHz) filtering capability for $\theta <6^{{}^{\circ}}\;$ with at least three noticeable band gap areas. It can be realized that there is a clear difference in the band gap areas by changing the arm angles from $\theta =+4^{{}^{\circ}}$ to $\theta =+6^{{}^{\circ}}$. The sensitivity of the band gap areas to the arm angle may be better realized by examination of the percentage of the bandgap coverage area for the passive MRE-based periodic structure versus arm angles, as presented in Figure 8b. Results clearly show that zero and low negative angles provide relatively high percentage bandgap areas compared with positive angles. For the sake of simplicity and ease of future manufacturing, the angle of θ=0 may be selected.

#### 3.1.3. Effect of Iron Volume Fractions on Passive MRE Unit Cell

In this section, the proposed MRE-based modified periodic structure with a cylindrical core radius of L/R=5, and the arm angle of *θ* = 0 is considered to evaluate the effect of various iron volume fractions on the wave propagation analysis. Material properties of MREs with combinations of the non-magnetic elastomer (silicon rubber) and different percentages of iron volume fractions are extracted from experimental data reported by Kyung Hoon Lee et al. [54] and presented in Table 1. Results are shown in Figure 9. Results show that changing the iron volume fraction of the passive periodic unit cell (field-off) does not noticeably change the wave propagation behaviour of the MRE unit cell. This is in part due to the fact that both the elastic modulus and density of the magnetorheological elastomer change proportionally by changing the percentage of iron volume, which yield no change in local resonances.

It is noted that the percentage of iron volume fraction would substantially change wave propagation properties of the proposed MRE-based periodic structures in the presence of the applied magnetic field, as will be discussed in the following section. For this purpose, the proposed modified periodic unit cell with two ratios *L*/*R* = 5 and 8 and angles $\theta =0\;\mathrm{and}\;-1^{{}^{\circ}}$ made of MREs with different percentages of volume fractions (i.e., 6.08%, 13.94%, and 18.48%) will be subjected to the varied applied magnetic field to investigate its bandgap tunability.

### 3.2. Adaptive Feature of Proposed Architected Structures

The modulus of the magnetorheological elastomers (MREs) can be adaptively varied through variation of an external magnetic field. This unique adaptive behaviour can be better realized in Figure 10, which shows the variation of Young’s modulus of MREs with respect to the applied magnetic flux density for various iron volume fractions provided in Table 1. As it can be realized, the difference between Young’s Modulus under lower and higher magnetic fields increases by increasing the iron volume fraction. In other words, the dependency of Young’s Modulus on the magnetic field in MREs with higher volume fractions is more pronounced.

Here, wave propagation analysis has been conducted for the MRE-based unit cell with two combinations of geometrical parameters (L/R=5, θ=0 and L/R=8, $\theta =-1^{{}^{\circ}}$) and three iron volume fractions (i.e., 6.08%, 13.94%, and 18.48%) in order to investigate the variation of the bandgap areas under varied external magnetic flux density, B, in the range of 0 to 250 mT. In the first part of this section, the periodic unit cell with L/R=5 and θ=0 is considered, and the branch diagrams for three different iron volume fractions and two magnetic fields (B=75 mT and B=225 mT) are obtained and shown in Figure 11. The lower and upper branches of the first four bandgap areas associated with the 5th and 6th, 7th and 8th, 9th and 10th, and 11th and 12th,^,^ as listed in Table 2, are shown in Figure 10 with different colours. This helps to observe the variation of these bandgaps under the different volume of iron fractions and applied magnetic flux densities.

The purple colour branches associated with 5th and 6th mode shapes are related to the first band gap. It can be realized that the upper branch of this gap (6th mode shape) is a constant line representing local resonance, irrespective of the amount of volume fraction of magnetic particles and applied magnetic flux densities. The first band gap is beneficial because it is in the range of low frequency. For instance, under B=75 mT, the first band gap changes from 700.4 Hz–1109.8 Hz for MRE with ϕ=6.08% to 711.5 Hz–1125.2 Hz for MRE with ϕ=13.94%. The band gap can also be varied in real time through the application of an external magnetic field. For instance, the first band gap for MRE with ϕ=18.48% changes from 776.9 Hz–1214.9 Hz under B=75 mT to 1230.7 Hz–1701.9 Hz under B=225 mT. As illustrated in the band diagrams in Figure 11, increasing the magnetic flux density increases the branch frequencies related to each bandgap. This is mainly due to the magnetic field stiffening effect of MREs, as also reported in previous studies [57]. For instance, for ϕ=6.08% (Figure 11a,b), the start point of the seventh branch of the band diagram (coloured red after purple) in high symmetry points of G (free body motion) increases from ω=1308.7 Hz under B=75 mT to ω=1838.2 Hz under B=225 mT (almost 40% increase), whereas that for the eighth branch of the band diagram (coloured red before green) only changed from ω=1966.4 Hz under B=75 mT to ω=2141.6 Hz under B=225 mT, which causes the width of the band gap to gradually reduce between the seventh and eighth branches by increasing the magnetic field. 

This unequal variation in frequencies associated with different mode shapes under various magnetic fields is not always the cause of the elimination of band gaps in periodic structures. For instance, the 9th and 10th branches of the band diagrams (coloured green) generate a stop band area by increasing the magnetic field, as shown in Figure 11c,d (ϕ=13.94%). It can be realized that the start point of these branches varies from 1988.6 Hz and 2308.3 Hz to 2182.1 Hz and 2591.6 Hz, respectively, by increasing the magnetic flux density from B=75 mT to B=225 mT.

The possibility of changing the stop band areas under different magnetic fields provides a unique opportunity to design and develop a tunable periodic structure using magnetic stimuli. To further examine this unique feature of the proposed MRE periodic unit cell, the band gap frequency variation with respect to the applied magnetic flux density has been evaluated for MREs with various iron volume fractions, as illustrated in Figure 12. As it can be realized, wide variation in bandgap areas can be achieved through the application of the magnetic field without changing any geometrical parameters. This is due to the unique functionality of the MRE materials, which their elastic modulus can be substantially varied using an applied magnetic field, which subsequently affects the operating frequencies of the periodic structures [37,58].

As illustrated in Figure 12, MRE-based periodic structures with L/R=5 and θ=0 and various iron volume fraction have low-frequency filtering areas with barely wide band gaps higher than ω>3 kH. To investigate the behaviour of MRE-based periodic structure with bandgaps at higher frequencies, the wave propagation analysis has also been conducted on an MRE unit cell with the cylindrical core radius of L/R=8 and the arm angle of $\theta =-1^{{}^{\circ}}$. Results for band diagrams are depicted in Figure 13. It is noted that this periodic structure has low-, mid-, and high-frequency band gaps, which are easily tunable through variations of the magnetic field. Similar to Figure 11, the specific branches of band structures are marked with different colours in order to study their variation under various iron volume fractions and applied magnetic fields. For instance, in the design with an iron volume fraction of ϕ=6.08%, the first band gap area (between two purpole branches) is easily tunable and can be eliminated by increasing the magnetic flux. Additionally, this specified configuration of the unit cell is able to have more band gap areas under low magnetic flux densities. As depicted in Figure 13a,c,e, under B=75 mT, there are six stop band areas, regardless of the volume of iron fractions. These bandgap areas are listed in Table 3:

Similar to the previous configuration (L/R=5 and θ=0), results for the variation of the bandgap frequency areas with respect to the applied magnetic flux density for the current configuration (L/R=8 & $\theta =-1^{{}^{\circ}}$) are shown in Figure 14. Examination of results reveals that this configuration shows nearly the same bandgap behaviour for MREs with ϕ=6.08% and ϕ=13.94%, although the continuity of the band gaps for the higher iron volume fractions is more than that of lower iron volume fraction. Although the lower volume fractions have the same behaviour, and it is possible to use them in some band gap areas, as shown in Figure 14, this configuration depicts better performance with higher volume fractions. As shown in Figure 14c, this unit cell with a volume fraction of ϕ=18.48% provide wide band gap in low frequencies (available until B=250 mT) and it can cover more areas in higher frequencies above 4.5 kHz compared with those with lower iron volume fraction.

To further clarify this, the variability of bandgap percentage areas for the proposed configuration of the periodic unit cell, featuring MREs with different iron volume fractions, with respect to the applied magnetic flux density is analyzed and shown in Figure 15. Results in Figure 15a show that the MRE-based periodic structure with a cylindrical core of L/R=5 and trapezoidal arm angle of θ=0 shows impressive results under magnetic flux density in the range of 50 mT to 100 mT and have a maximum band gap percentage of nearly 35% using high iron volume fraction (i.e., ϕ=18.48%). It is interesting to note that after 100 mT, the percentage of the bandgap area drastically reduces, reaching around 20% under 250 mT, regardless of the amount of iron volume fraction used. Moreover, further examination of results reveals that, although the percentage of bandgap areas for MRE-based periodic unit cells with ϕ=6.08% and ϕ=13.94% are nearly the same under magnetic flux density of 100 mT, the unit cell with ϕ=13.94% shows more reduction in bandgap areas compared with the unit cell with ϕ=6.08% for flux density above 100 mT. This is in part due to the fact that the third band gap in Figure 12a (between black branches in Figure 11a,b) is wider and has more filtering properties in high magnetic fields. 

The results for the other configuration of the periodic unit cell (L/R=8 and $\theta =-1^{{}^{\circ}}$) are also shown in Figure 15b. Again, results confirm a higher bandgap area for the MRE unit cell with a high-volume fraction (ϕ=18.48%) through the entire range of the induced magnetic flux density. MRE unit cells with iron volume fractions of 6.08% and 13.94% almost show the same percentage of bandgap area through the entire magnetic flux range, as confirmed before. In contrast to the MRE unit cell with geometrical parameters of L/R=5 and θ=0, the MRE unit cell with L/R=8 and $\theta =-1^{{}^{\circ}}$ shows a higher percentage of bandgap area in the absence of the magnetic field. Based on the results shown in Figure 15b, the bandgap percentage area initially declines as the flux density increases up to nearly 75 mT, then increases by increasing the flux density reaching a peak value around 150 mT and then again declines for flux densities above 150 mT.

## 4. Conclusions

In the present manuscript, the wave propagation characteristics of magnetoactive periodic structures featuring magnetorheological elastomers (MREs) in the absence and presence of the applied magnetic stimuli were studied. A conventional model of periodic structures (square model), which does not possess noise filtering properties, was initially selected, and, subsequently, a passive attenuation zone was proposed by adding a cylindrical core to the center of the basic square lattice and changing its cross-shaped arm angles. By introducing the added cylinder and angle changes, the mass distribution in the unit cell was altered, and the model became close to the mass-spring model, which assisted the periodic structure in having band gap areas. Performance analysis was then conducted to investigate the effect of core cylinder radius and angle of cross arms as well as the percentage of iron volume fraction in MRE on the bandgap frequency ranges. It was concluded that better wave-filtering performance may be attained by introducing a large radius of the cylindrical core and/or negative trapezoidal angle of arms. The developed configurations of the unit cell were considered the basic models for the development of adaptive MRE-based metamaterial. The finite element model of the adaptive MRE-based architected structures was designed and developed using COMSOL Multiphysics. Bloch theorem was used to analyze bandgap frequency regions in reciprocal space. Finally, the effect of the applied magnetic field on the variation of the bandgap frequency ranges, as well the percentage of bandgap areas, was studied for MRE-based unit cells with two configurations capable of attenuating low-, mid-, and high-frequency vibrations and noise in the frequency range of 0 to 5 kHz. Results clearly show that proposed MRE-based periodic structures can effectively generate new bandgap regions through the application of the magnetic field. This can provide a unique opportunity to filter unwanted vibration and noise with time-varying frequency components.

## Figures and Tables

**Figure 1 polymers-15-00735-f001:**
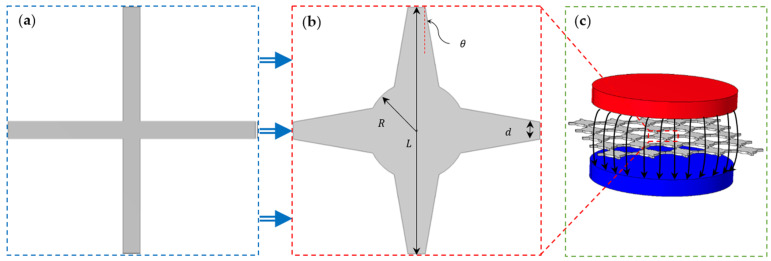
Schematic of in-plane periodic architected metamaterial: (**a**) the conventional square unit cell; (**b**) the square unit cell with cylindrical core and trapezoidal arm; (**c**) periodic structure affected by magnetic fields.

**Figure 2 polymers-15-00735-f002:**
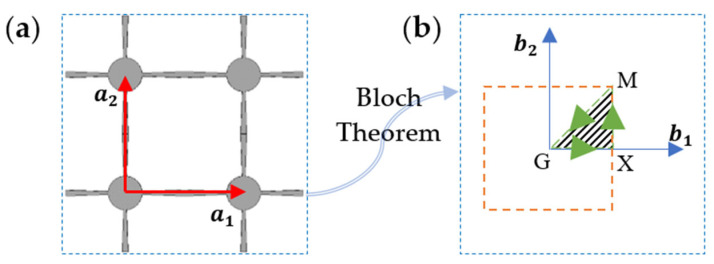
Schematic of in-plane periodic architected metamaterial: (**a**) a periodic sample structure and its direct unit cell vectors; (**b**) FBZ (orange dotted empty square) and IBZ (green dotted hatched triangular area) of the square unit cell.

**Figure 3 polymers-15-00735-f003:**
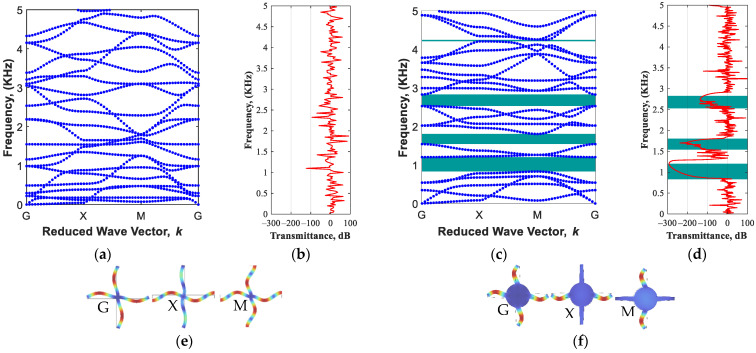
Initial passive design: (**a**) dispersion curves, (**b**) transmittance of conventional periodic structure based on a square lattice (Figure 1a); (**c**) dispersion curves, (**d**) transmittance of optimized periodic structure with design parameters of L/R=5 and $\theta =3^{{}^{\circ}}$ (Figure 1b); (**e**) the sixth mode shapes of the conventional lattice in various high symmetry points; (**f**) the sixth mode shapes of the modified lattice in various high symmetry points, MRE with 13.94% volume fraction.

**Figure 4 polymers-15-00735-f004:**
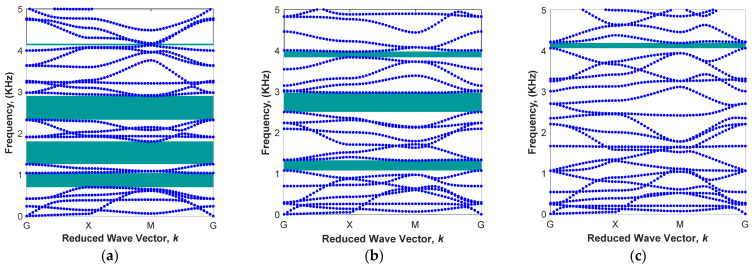
Various band structures in different radiuses of cylindrical core in the fixed arm angle θ=0 and iron volume fraction of 13.94%: (**a**) L/R=5; (**b**) L/R=8; (**c**) L/R=13.

**Figure 5 polymers-15-00735-f005:**
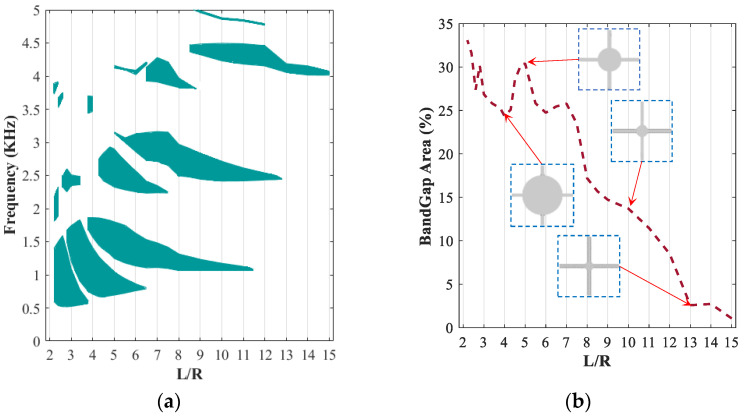
Evolution of bandgap areas in a different radius of cylindrical core (2.1<L/R<15) in the fix angle of θ=0 and Iron volume fraction of 13.94%: (**a**) bandgap area; (**b**) percent of bandgap area in the range of 0 and 5 kHz.

**Figure 6 polymers-15-00735-f006:**
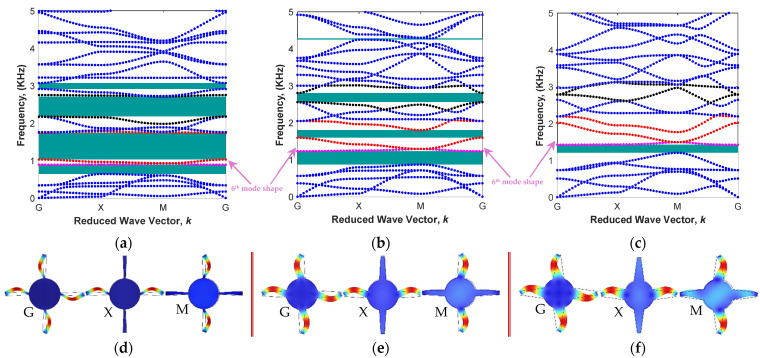
Various band structures in different arm angles with the fixed cylindrical core radius of L/R=5 and MRE with iron volume fraction of 13.94%: (**a**,**d**) $\theta =-1.5^{{}^{\circ}}$; (**b**,**e**)$\;\theta =+3.5^{{}^{\circ}}$; (**c**,**f**) $\theta =+8^{{}^{\circ}}$, as well as the sixth mode shapes of the structures in high symmetry points (Sixth mode shape colored in purple represents the first band gap; Seventh and eighth mode shapes colored in red and represent the second band gap; Eleventh and twelfth mode shapes colored in black and represent the third band gap).

**Figure 7 polymers-15-00735-f007:**
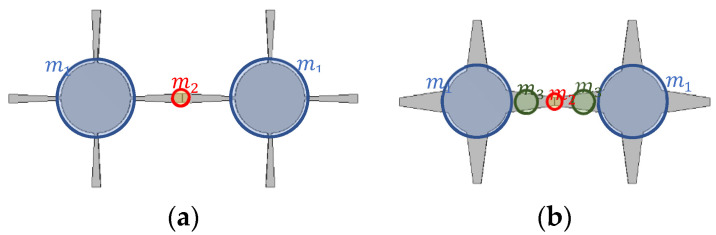
The distribution of mass along the periodic structure with various angles of trapezoidal arms: (**a**) $\theta =-1.5^{{}^{\circ}}$; (**b**) $\theta =+8^{{}^{\circ}}$.

**Figure 8 polymers-15-00735-f008:**
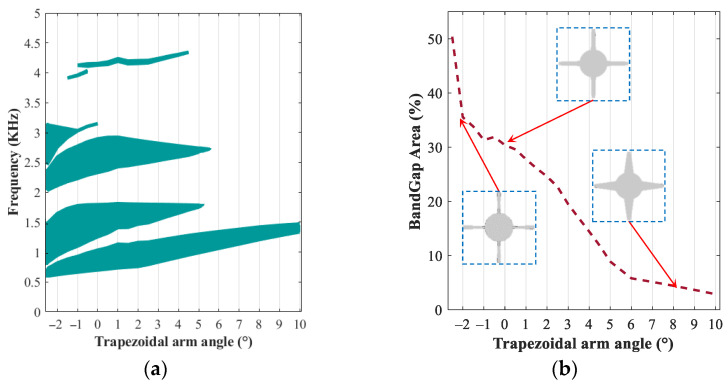
Evolution of bandgap areas in different arm angles (−2.5<θ<10) in the fixed cylindrical core radius of L/R=5 and iron volume fraction of 13.94%: (**a**) bandgap area; (**b**) percent of bandgap area in the range of 0 and 5 kHz.

**Figure 9 polymers-15-00735-f009:**
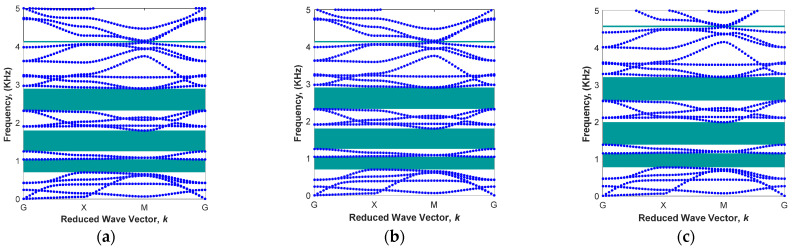
Various band structures in different iron volume fractions with the fixed cylindrical core radius of L/R=5 and the fixed arm angle θ=0: (**a**) ϕ=6.08%; (**b**) ϕ=11.47%; (**c**) ϕ=18.48%.

**Figure 10 polymers-15-00735-f010:**
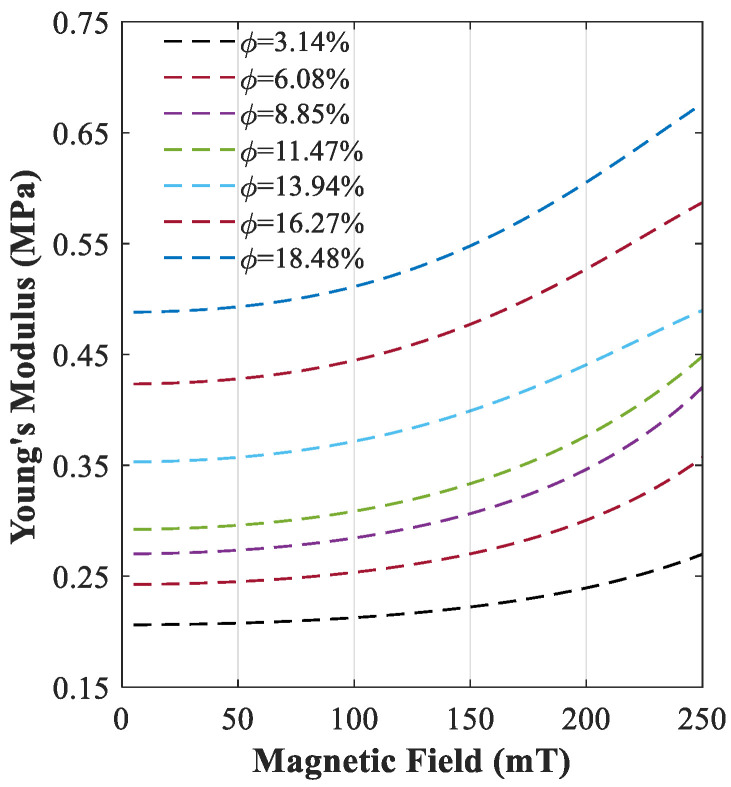
Dependence of material properties (Young’s Modulus) to the magnetic field for various iron volume fractions.

**Figure 11 polymers-15-00735-f011:**
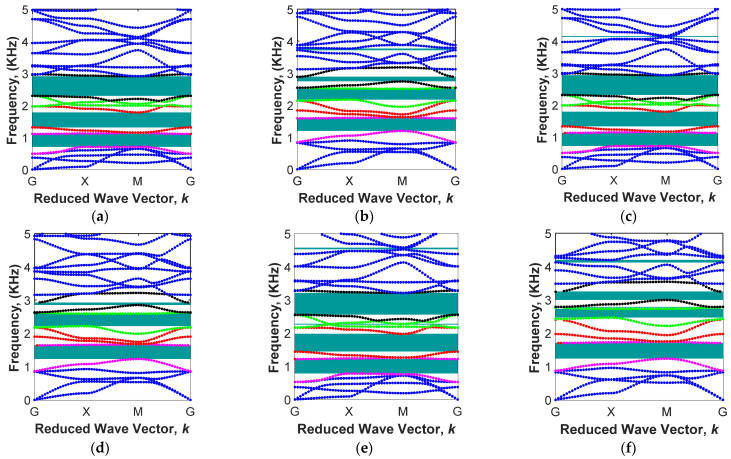
Active band structures in different magnetic fields in various volume fractions with the fixed cylindrical core radius of L/R=5 and the fixed arm angle θ=0: (**a**) ϕ=6.08% & B=75 mT; (**b**) *ϕ* = 6.08% & B=225 mT; (**c**) ϕ=13.94% & B=75 mT; (**d**) ϕ=13.94% & B=225 mT; (**e**) ϕ=18.48% & B=75 mT; (**f**) ϕ=18.48% and B=225 mT (Fifth and Sixth mode shapes colored in purple and represent the first band gap; Seventh and eighth mode shapes colored in red and represent the second band gap; Ninth and tenth mode shapes colored in green and represent the third band gap; Eleventh and twelfth mode shapes colored in black and represent the fourth band gap).

**Figure 12 polymers-15-00735-f012:**
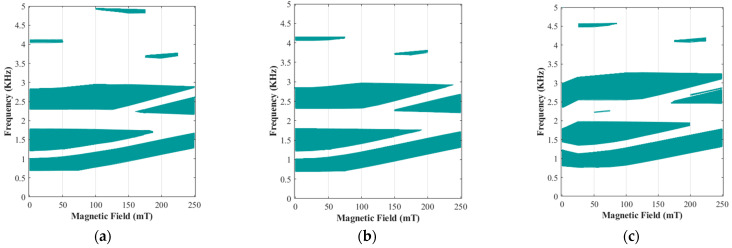
Evolution of active bandgap areas in different magnetic fields (0<B<250 mT) with the various iron volume fraction, fixed cylindrical core radius of L/R=5, and the fixed arm angle θ=0: (**a**) ϕ=6.08%; (**b**) ϕ=13.94%; (**c**) ϕ=18.48%.

**Figure 13 polymers-15-00735-f013:**
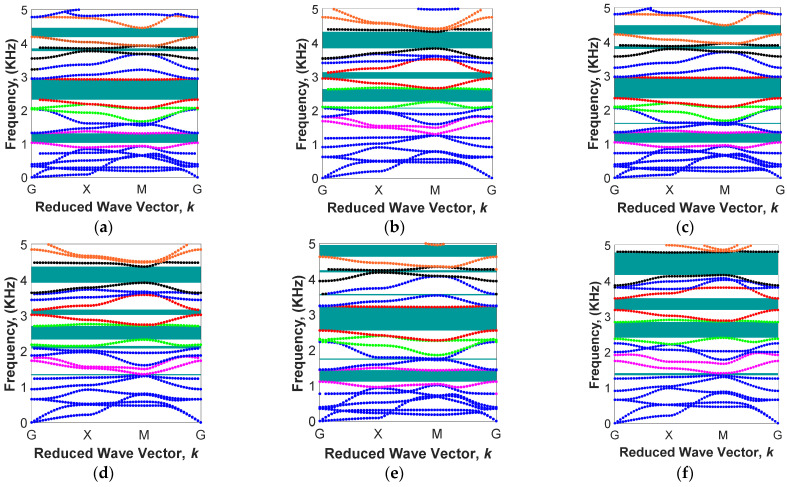
Active band structures in different magnetic fields in various volume fractions with the fixed cylindrical core radius of L/R=8, and the fixed arm angle $\theta =-1^{{}^{\circ}}$: (**a**) ϕ=6.08% & B=75 mT; (**b**) ϕ=6.08% & B=225 mT; (**c**) ϕ=13.94% & B=75 mT; (**d**) ϕ=13.94% & B=225 mT; (**e**) ϕ=18.48% & B=75 mT; (**f**) ϕ=18.48% and B=225 mT (Seventh and eighth mode shapes colored in purple and represent the first band gap; Eleventh and twelfth mode shapes colored in green and represent the second band gap; thirteenth and fourteenth mode shapes colored in red and represent the third band gap; Seventeenth and eighteenth mode shapes colored in black and represent the fourth band gap; Nineteenth and twentieth mode shapes colored in orange and represent the fifth band gap).

**Figure 14 polymers-15-00735-f014:**
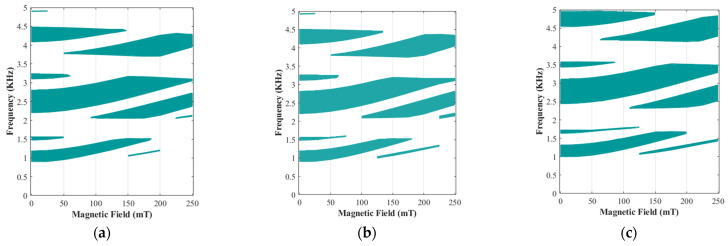
Evolution of active bandgap areas in different magnetic fields (0<B<250 mT) with the various iron volume fraction, fixed cylindrical core radius of L/R=8, and the fixed arm angle $\theta =-1^{{}^{\circ}}$: (**a**) ϕ=6.08%; (**b**) ϕ=13.94%; (**c**) ϕ=18.48%.

**Figure 15 polymers-15-00735-f015:**
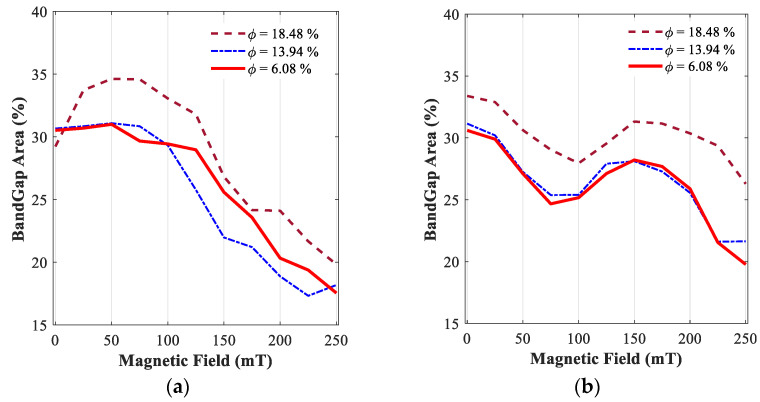
Evolution of active bandgap areas percent in the range of 0 and 5 kHz and different magnetic fields (0<B<250 mT) with various iron volume fractions, cylindrical core radius, and arm angles: (**a**) L/R=5 & θ=0; (**b**) L/R=8 & $\theta =-1^{{}^{\circ}}$.

**Table 1 polymers-15-00735-t001:** Experimentally-measured material properties of magnetorheological elastomers with different volume fractions (Poisson’s ratio as 0.48) [54].

Iron Volume Fraction (%)	Young’s Modulus of Filled Elastomer (MPa)	Density of Filled Elastomer $\left(kg/m^{3}\right)$	Effective Magnetic Susceptibility Difference Δχ
3.14	0.22661	1019	0.169
6.08	0.26147	1205	0.457
8.85	0.28785	1307	0.953
11.47	0.30889	1413	2.03
13.94	0.36696	1675	6.02
16.27	0.43402	1770	8.34
18.48	0.49576	1860	11.53

**Table 2 polymers-15-00735-t002:** Branch numbers related to various bandgaps in band diagrams of the modified unit cell with a cylindrical core of L/R=5 and the arm angle of θ=0.

Order of Bandgap	Lower Branch (nth Mode Shape)	Upper Branch (nth Mode Shape)	Colour
First Bandgap	5th	6th	Purple
Second Bandgap	7th	8th	Red
Third Bandgap	9th	10th	Green
Fourth Bandgap	11th	12th	Black

**Table 3 polymers-15-00735-t003:** Band gap areas for the modified unit cell with a cylindrical core of L/R=8, and the arm angle of $\theta =-1^{{}^{\circ}}$ in B=75 mT.

Iron Volume Fraction (%)	6.08	13.94	18.48
1st band gap (kHz)	1.02–1.31	1.03–1.36	1.10–1.43
2nd band gap (kHz)	No Band Gap	1.58–1.61	1.72–1.76
3rd band gap (kHz)	2.31–2.91	2.33–2.94	2.54–3.21
4th band gap (kHz)	No Band Gap	No Band Gap	3.53–3.53
5th band gap (kHz)	3.76–3.85	3.78–3.87	4.18–4.24
6th band gap (kHz)	4.17–4.46	4.20–4.49	4.62–4.96

## Data Availability

Data will be available upon request.

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
