# Peer review of "Analysis of an Adaptive Periodic Low-Frequency Wave Filter Featuring Magnetorheological Elastomers"

_polymers, 2023, doi:10.3390/polym15030735_

Round 1

Reviewer 1 Report

In this paper, a two-dimensional phononic crystal is designed using a magnetorheological elastomers to tune the band gap by magnetic field. The band structures are simulated using finite element model implemented in COMSOL. In my opinion, this is an interesting work, and it is worth studying deeply, but before it is accepted, there are still some problems that need to be explained and revised by the author.

1.      In fact, there has been some research on the construction of tunable phononic crystals using soft materials. Such as,

(a)   ‘’Tunability of band gaps of programmable hard-magnetic soft material phononic crystals’’ Acta Mechanica Solida Sinica, 35(5), pp.719-732.

(b)   "Functionally Graded Soft Dielectric Elastomer Phononic Crystals: Finite Deformation, electro-elastic longitudinal waves, and band gaps tunability via electro-mechanical loading" International Journal of Applied Mechanics, 14(5), 2250050 (2022).

(c)   "Gradient-based topology optimization of soft dielectrics as tunable phononic crystals" Composite Structures 280, 114846 (2022).

   The author should comment and cite in the reference list.

2.      Authors should provide details about implementation of the periodic boundary conditions in the FE formulation for extracting band structures.

3.      Please supplement the finite-period transmission spectrum to verify the band gap.

4.      Improve the quality of the figures.

Author Response

We greatly appreciate the attempt to review our manuscript and provide valuable comments which have certainly helped us to improve the quality of our presentation. The point-by-point responses are provided below, and all modifications in the original manuscript have been highlighted in blue.

Comments #1:      In fact, there has been some research on the construction of tunable phononic crystals using soft materials. Such as,

(a) ‘’Tunability of band gaps of programmable hard-magnetic soft material phononic crystals’’ Acta Mechanica Solida Sinica, 35(5), pp.719-732.

(b)   "Functionally Graded Soft Dielectric Elastomer Phononic Crystals: Finite Deformation, electro-elastic longitudinal waves, and band gaps tunability via electro-mechanical loading" International Journal of Applied Mechanics, 14(5), 2250050 (2022).

(c)   "Gradient-based topology optimization of soft dielectrics as tunable phononic crystals" Composite Structures 280, 114846 (2022).

The author should comment and cite in the reference list.

Response: Thanks for your comment. All these papers and some other new papers are commented in the introduction section and added to the reference list.

Comments #2:      Authors should provide details about the implementation of the periodic boundary conditions in the FE formulation for extracting band structures.

Response: Thanks for your comment. This expression was missed in our manuscript. We added that to the context and highlighted it in the revised manuscript on page 6.

Comment #3:      Please supplement the finite-period transmission spectrum to verify the band gap.

Response: Thanks for this pertinent remark. In order to verify the band gap, the finite-period transmission spectrum is added to Fig. 3 on page 8 and related description is provided on page 7

Comment # 4:      Improve the quality of the figures.

Response: Thanks. The quality of the Figures are improved in the revised manuscript.  

Reviewer 2 Report

Report attached

Author Response

We greatly appreciate the attempt to review our manuscript and provide valuable comments which have certainly helped us to improve the quality of our presentation. The point-by-point responses are provided below, and all modifications in the original manuscript have been highlighted in blue.

This study aims to investigate the Bandgaps of elastic waves in periodic structures made of magnetorheological elastomers (MREs) by adding a cylindrical core mass to the center of the conventional geometry and changing arm angles. This modification creates new bandgap areas and improves wave-filtering performance. The study also explores the use of MREs to create magnetoactive metamaterials which can adaptively change and tune bandgap areas using an external magnetic field. The study uses the finite element model of the proposed MRE-based periodic unit cell and the Bloch theorem to investigate the ability of the proposed adaptive structure to filter low-frequency noise and vibration in a wide frequency band. The findings can provide guidance for the development of novel adaptive periodic structures for noise filtering.

The paper can improve the following aspects:

Comments #1: As a minor point, the notation is not consistent: sometimes, the authors use bold (for vectors and tensors), but very frequently switch to using […] for matrix-like notation, or using an upper arrow for vectors. That would be 3 different notation schemes. It is suggested to unify the notation for better readability of the paper.

Response: Thanks for your comment and apologize for the confusion. We have fixed this issue (vectors have an upper arrow and matrices an upper bar). All changes are highlighted in the revised manuscript on pages 4, 5 and 6.

Comments #2: The paper mentions the use of the hyperelastic neo-Hookean model. Do the authors consider large deformations and associated geometrical and material nonlinearity? This aspect is not clear, as it seems that the governing equations (5) only apply to small amplitude linear waves. This requires clarification.

Response: Thanks for this pertinent remark. We have assumed a linear elastic wave motion. We have now clarified this issue on page 5 of the revised manuscript.

Comments #3:  Also, the authors mention incompressible model, but the finite element method cannot handle ideal incompressible materials. The authors should clarify how this assumption is made. As a result, it is important that the authors clarify that they consider only transverse (or shear) waves. Otherwise, there may be some confusion, as, for example, the keywords mention "sound filers" is typically associated with pressure /longitudinal waves.

Response: Thank you for your constructive comment. The model is considered nearly incompressible, and this is corrected in the manuscript (page 5). The transverse wave is correct, and the keyword on the first page is changed to wave filters.

Comments #4: On the other hand, the dependence of Poisson's ratio on the magnetic field is mentioned later in the texts; this creates further confusion if the incompressible or compressible material is considered, and hence, whether transverse or longitudinal waves are examined.  

Response: As mentioned in response to previous comment, the model is nearly incompressible and the transverse wave is considered in this context. The Poisson's ratio is considered to be 0.48 for all magnetorheological elastomers with different volume fractions based on Ref. [54] in the text.

Comments #5: Th authors mentioned that they use an experimentally measured dependence of material properties on a magnetic field. It is worth including that dependence plot in the main body of the paper.

Response: Thanks. To demonstrate the adaptive feature of MREs, the variation of modulus of MREs  with respect to the applied magnetic flux density, for different volume fraction of iron particles presented in Table 1, has been evaluated and shown in new Figure 10 on page 12.

Comments #6: The results in Fig. 5 seem somewhat not smooth. The authors are suggested to check the accuracy of the results or explain the irregularity and sharp changes in the behaviour, if possible.

Response: Thanks.  Results in Fig. 5 are double-checked and are correct. The sharp change in behaviour around L/R=5  is due to maximum change in the bandgap area in this radius (the bandgap area is 2.3<ω<2.9 kHz  which is between the 11th and 12th branches). This has been clarified in the paragraph before Figure 5.

Comments #7: Is there a switch from the local resonance band gap mechanism to Bragg scattering? If so, can it be illustrated

Response: Thanks. There was no switch between the local resonance bang gap mechanism to Bragg scattering.

Comments #8: The authors are encouraged to discuss the effect of the magnetic field, including magnetic field-induced stiffening, and magnetic field-induced deformation on the band gaps.

Response: Thanks. The effect of the magnetic field on variation of  the stiffness of MREs with different level of iron volume fractions and also its effect on the variation of the bandgap areas are thoroughly discussed in section 3.2 and  Figures 10 -15.

Comments #9: Since the considered setting is in 3D (based on the setting in Fig.1c), the out-of-plane effects need to be considered. For example, how the structure thickness would affect the band gap, would the magnetic stiffening and deformation be the same in out-of-plane and in-plane? etc.

Response: Thanks for your comment. The setting shown in Fig. 1c has periodic boundary conditions in 2 directions (X, Y), and the in-plane waves (x-y) are considered in this study. However, we will consider the effect of magnetic stiffening and out-of-plane thickness on out-of-plane waves in future investigations. 

Comments #10: The literature review is incomplete and misses many relevant works on elastic waves in magnetoactive composites. For example, a recent work of magnetoactive periodic metamaterial capable of tunable band gaps by a magnetic field is relevant [Ext Mech Lett 59:101957 2023]. There is also literature on elastic wave (transverse waves) band gaps in periodic layered MREs (or MAE laminates) under a magnetic field.

Response: Thanks for the suggestion. The literature review is updated. Also,  the recently published paper [Ext Mech Lett 59:101957 2023] and another related material on MAE laminates are now cited in the revised manuscript. Please see page 3 in the revised manuscript.

Reviewer 3 Report

The article is distinguished by the originality of the selected topic, structured correctly, and written in Standard English. The manuscript presents the enhanced and tune wave-propagation properties (Bandgaps) of periodic structures featuring magnetorheological elastomers (MREs).  The basic model of periodic structures (square unit cell with cross-shaped arms), which does not possess noise filtering properties in the conventional configuration, is considered. A passive attenuation zone is then proposed by adding a cylindrical core mass to the center of the traditional geometry and changing arm angles, which permitted to have new bandgap areas.

The importance of the study:

·         better wave-filtering performance may be achieved by introducing a large radius of the cylindrical core and low negative cross-arm angles.

·         The modified configuration of the unit cell was subsequently utilized as the basic model for the development of magnetoactive metamaterial using MRE capable of varying the bandgaps areas upon application of an external magnetic field.

·          The finite element model of the proposed MRE-based periodic unit cell was developed, and the Bloch theorem was employed to systematically investigate the ability of the proposed adaptive periotic structure to attenuate low-frequency noise and vibration.

·         the proposed MRE-based periodic wave filter can provide wide bandgap areas which can be adaptively changed and tuned using the applied magnetic field.

·         novel adaptive periodic structures to filter low-frequency noises in the wide frequency band.

 The introduction, methods, and results are presented correctly and a logical relationship between them is clearly observed.

In an initial review of the article, there were stylistic and grammatical mistakes that I hope the authors will avoid after the revision.

Suggest:

1.      Most of the literature used in the article is from the last 3 years! Part of the references is not according to the journal requirements.

2. The tables and graph design are made well!

3. The theoretical model is well established.

4.      Тhe discussion part is well written.

Author Response

Thanks for your very positive and constructive comments. As requested, the reference list is updated and adjusted based on the journal requirements. The manuscript has also been edited to the best of our ability to be free from typos.